

# Conceptualising carbon cycling pathways across different land-use types based on rates and ages of soil-respired CO₂

Luisa I. Minich[1, 2], Dylan Geissbühler[3], Stefan Tobler[1], Annegret Udke[1,5], Alexander S. Brunmayr[2], Margaux Moreno Duborgel[1,2], Ciriaco McMackin[4], Lukas Wacker[5], Philip Gautschi[5], Negar Haghipour[2,5], Markus Egli[4], Ansgar Kahmen[6], Jens Leifeld[7], Timothy I. Eglinton[2], Frank Hagedorn[1]

[1] Swiss Federal Institute for Forest, Snow and Landscape Research (WSL), Switzerland
[2] Biogeoscience, Department of Earth and Planetary Sciences, ETH Zurich, Switzerland
[3] Laboratory for the Analysis of Radiocarbon with AMS, Department of Chemistry, Biochemistry and Pharmaceutical Sciences, University of Bern, Switzerland
[4] Geochronology, Department of Geography, University of Zurich, Switzerland
[5] Laboratory for Ion Beam Physics, Department of Physics, ETH Zurich, Switzerland
[6] Physiological Plant Ecology, Department of Environmental Sciences, University of Basel, Switzerland
[7] Climate and Agriculture Group, Agroscope, Switzerland

*Correspondence to*: Luisa I. Minich (luisa.minich@gmail.ch)

**Abstract.** Soil carbon dioxide ($CO_2$) efflux constitutes a major carbon (C) transfer from terrestrial ecosystems to the atmosphere, driven by numerous metabolic and allocation processes in the plant-soil system. Land use affects key components of C cycling pathways through vegetation type, C allocation, abiotic conditions, and management impacts on soil organic matter (SOM). However, systematic comparisons of these pathways among land uses remain scarce. We measured *in situ* respiration rates and C isotopic signatures ($^{14}$C, $^{13}$C) of soil-respired $CO_2$ and its autotrophic and heterotrophic sources during summer and winter at 16 sites across Switzerland, covering temperate and alpine grasslands, forests, croplands, and managed peatlands. Our findings revealed significant differences in the rates, ages, and sources of soil-respired $CO_2$ between land-use types, reflecting variations in C cycling dynamics. We propose that respiration rates and ages of soil-respired $CO_2$ serve as comprehensive indicators to categorize C cycling into:

1. **High-throughput systems** (temperate grasslands): High respiration rates of young (< 10 years) $CO_2$ in autotrophic and heterotrophic components reveal rapid C cycling.

2. **Retarding systems** (alpine grasslands): Young (< 10 years) *in situ* $CO_2$ fluxes and dominance of autotrophic sources, but slow C cycling through SOM mainly due to cooler climatic conditions.

3. **Preserving systems** (forests): Decadal-old $CO_2$ reflects a delayed C transfer of assimilates back to the atmosphere through soil respiration.

4. **Destabilized C-depleted systems** (croplands): Reduced C inputs and tillage lead to C depletion and to respiratory losses of older C (~ 650 years).

5. **Destabilized hotspots** (managed peatlands): Release of ancient C (~ 3000 years) due to disturbances in natural C cycling by drainage.

Our results suggest that the relationship between rates and ages of soil-respired $CO_2$ can serve as a robust indicator of C retention and destabilization along the trajectory from natural to anthropogenically disturbed systems on a global scale.





**Graphical abstract**

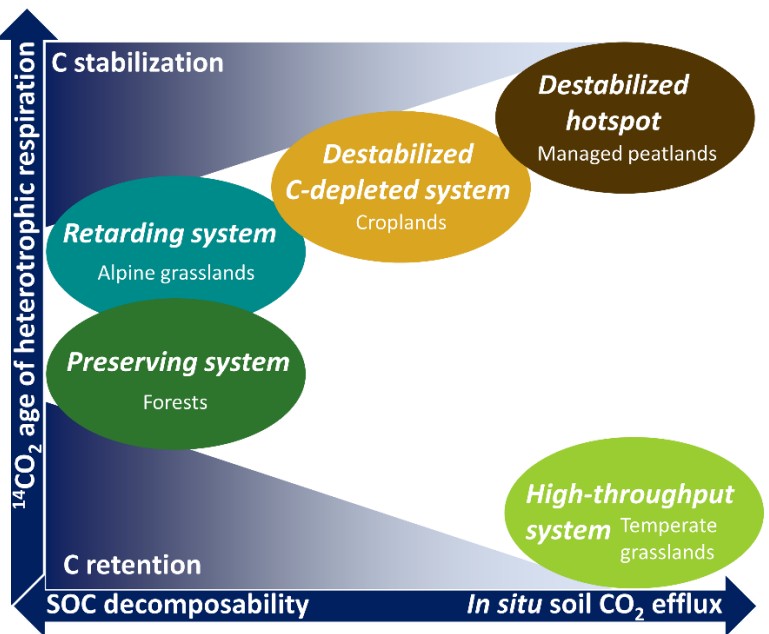

## 1 Introduction


Soil carbon dioxide ($CO_2$) efflux is one of the largest carbon (C) fluxes between terrestrial ecosystems and the atmosphere with $CO_2$ release from soils exceeding fossil fuel $CO_2$ emissions (IPCC, 2021; Nissan et al., 2023). Soil $CO_2$ fluxes (i.e., soil respiration) originate from root activity related to the metabolic processing and allocation of C in plants (i.e., autotrophic respiration), as well as the mineralization of various soil organic matter (SOM) compounds by microorganisms (i.e.,

heterotrophic respiration) (Kuzyakov, 2006; Trumbore, 2006). In this study, we refer to root respiration as all respiratory processes within the rhizosphere, including autotrophic respiration by root activity as well as heterotrophic respiration related to the consumption of root exudates by mycorrhiza and microorganisms (Hanson et al., 2000; Trumbore, 2006). Relative source contributions are essential to determine the pathways and velocity of belowground C cycling. A dominance of autotrophic respiration indicates high gross primary productivity (GPP) (Schulze et al., 2009) or rapid cycling of plant

assimilates through the plant-soil system (Diao et al., 2022; Fuchslueger et al., 2016; Gao et al., 2021), largely bypassing storage in soil. In contrast, a higher contribution of heterotrophic respiration indicates that a large fraction of C enters the soils as plant detritus, which is then respired during degradation and transformation processes.

Land use impacts soil respiration and its source contributions through various factors such as vegetation type, root density, nutrient input, and management practices, altering soil structure and C cycling pathways within the plant-soil system (e.g.,

Oertel et al., 2016; Rong et al., 2015; Schaufler et al., 2010; Xiao et al., 2021). In Europe, grasslands have the highest soil



respiration rates, followed by wetlands, croplands, and forests (Schaufler et al., 2010). Grasslands are among the most productive land-use types in Europe, due to their high GPP and net primary productivity (NPP) (Schulze et al., 2009), high belowground C allocation (Fuchslueger et al., 2016; Hagedorn & Joos, 2014; R. Wang et al., 2021), dense rooting system, and rapid fine root turnover (Leifeld et al., 2015; Solly et al., 2013) which together accelerate C cycling through the plant-soil

system. In forests, aboveground C inputs are greater than in agricultural systems, where aboveground biomass is regularly removed (Hiltbrunner et al., 2013; Keel et al., 2019). These inputs – particularly woody debris and conifer litter – decompose slowly and lead to the formation of organic layers (Hiltbrunner et al., 2013; Zanella et al., 2011), reducing the rate of C release back to the atmosphere. In arable land used for crop production, tillage and biomass removal disturb natural C cycling and induces the depletion of SOC stocks (Keel et al., 2019), especially in managed C-rich peatlands (Leifeld et al., 2005). Swiss

peatlands, which store large amounts of C on a per area basis, have been intensively used for agriculture over the past century. Conversion of the majority (82 %) of these peatlands to other land uses has led to a severe degradation, causing high C losses (Wüst-Galley et al., 2020).

Despite their relevance in the global C cycle, soil $CO_2$ effluxes and their sources remain one of the greatest sources of uncertainty in global C budgets and in understanding climate feedbacks (Konings et al., 2019; Li et al., 2016; Tharammal et

al., 2019). One reason is the methodological challenge of quantifying the sources of soil-respired $CO_2$ under natural conditions. Common approaches rely on destructive techniques such as trenching or girdling (e.g., Diao et al., 2022; Shi et al., 2022; Wunderlich & Borken, 2012), are constrained by their short-term nature such as $^{13}C$ pulse-labelling (Gao et al., 2021; R. Wang et al., 2021), or by their differences in natural abundance $\delta^{13}C$ signatures which do not allow for distinction of autotrophic sources from the decomposition of young SOC compounds (e.g., Diao et al., 2022; Millard et al., 2010). Radiocarbon ($^{14}C$)

analysis provides a powerful tool to determine the age of respired $CO_2$, which in turn corresponds to the mean transit time of C – the time C spends in the terrestrial ecosystem from photosynthesis until respiration (Sierra et al., 2017). Nuclear weapons testing in the 1950s and 1960s led to nearly a doubling in the amount of atmospheric $^{14}CO_2$ in the Northern hemisphere (commonly known as the "bomb-spike"), inadvertently serving as a large-scale tracer experiment. Since then, atmospheric $^{14}CO_2$ levels have been diluted due to the emission of $^{14}C$-free fossil fuels (Schuur et al., 2016) and the incorporation of $^{14}CO_2$

into the ocean and terrestrial ecosystems. The $^{14}C$ bomb-spike can be used to investigate C incorporation and cycling in terrestrial ecosystems on decadal time scales (Graven et al., 2024). In combination with stable carbon ($^{13}C$) isotope analysis, $^{14}C$ measurements of $CO_2$ from *in situ* soil respiration as well as autotrophic and heterotrophic respiration from incubations provide a non-destructive approach to determine source contributions to total soil respiration based on their distinct isotopic signatures (e.g., Borken et al., 2006; Schuur & Trumbore, 2006). Soil incubation experiments under controlled conditions can

further inform about SOC decomposability, the SOC fraction that is potentially readily available for microbial decomposition, by relating rates of heterotrophic respiration to SOC contents of the soil samples (Schädel et al., 2019). High SOC decomposability suggests a large fraction of readily available organic matter, whereas low SOC decomposability indicates greater persistence and stability of SOC and/or advanced decomposition of more labile SOM.



Radiocarbon and stable isotopic approaches have been used to investigate the age and source contributions of respired $CO_2$ in
natural ecosystems (e.g., Hicks Pries et al., 2013; Schuur & Trumbore, 2006; Wunderlich & Borken, 2012). However, these
techniques have not yet been used in agro-ecosystems such as intensely managed grasslands, croplands, and drained peatlands.
In our study, we investigated the magnitude, velocity, and pathways of C cycling across five dominant land-use types in
Switzerland which span a gradient from natural to intensely managed and disturbed ecosystems: grasslands (temperate and
alpine), forests, croplands, and managed peatlands. We measured soil respiration rates and C isotopic signatures ($\Delta^{14}CO_2$ and
$\delta^{13}CO_2$) of total soil respiration and its sources and used Bayesian mixing models to determine the age and source contribution
for each land-use type. Our main goals were (i) to assess how the flux rates, ages, and source contributions of soil-respired
$CO_2$ vary across dominant land-use types in Switzerland, (ii) to determine how these variables change seasonally, and (iii) to
develop a conceptual framework to describe how $CO_2$ age and soil respiration rates can be used to identify and characterise C
cycling pathways in various land-use types, in order to detect SOC destabilization, and to assess the vulnerability of C cycling
in these systems to land-use and climate change.

We hypothesized that (i) temperate grasslands primarily respire young $CO_2$ at a high rate due to high belowground C allocation
and high autotrophic contributions driven by abundant fine root biomass; (ii) alpine grasslands respire older $CO_2$ because
colder climate slows down C turnover in the plant-soil system; (iii) forests respire older $CO_2$ at lower rates than grasslands,
due to slower litter decomposition and a higher proportion of heterotrophic respiration, (iv) croplands release old $CO_2$ at low
rates, because of reduced C inputs and tillage, which deplete soils in labile C and leave behind persistent, older SOC; and
finally (v) managed peatlands release the oldest $CO_2$ at high rates, driven by the decomposition of preserved C and high
heterotrophic respiration following drainage.

## 2 Materials and methods

### 2.1 Study sites

We sampled soil-respired $CO_2$, roots, and soil from 16 sites of five dominant land-use types in Switzerland: grasslands
(temperate and alpine), forests, croplands, and managed peatlands (drained and used as croplands). Land-use types were
chosen based on established classifications following the LULUCF categories defined by the IPCC (IPCC, 2021; IPCC,
2003), with an additional distinction between temperate and alpine grasslands to account for climatic differences. The
16 sites vary in their physico-chemical soil properties, span a climatic as well as elevational gradient from 393 to 2630 meters
above sea level (a.s.l.) and encompass four of Switzerland's five ecoregions (Fig. 1, Table S1). Site properties are typical for
the respective land-use type across Switzerland. Grasslands were either used as meadows (Chamau, Muldain), including
mowing and manure application, or as pasture (Jaun, alpine grasslands), including grazing. The three cropland sites resemble
each other in crop type (summer: maize, winter: wheat; Table S1) and management practices, and are part of long-term field
trials of the Swiss federal research institute Agroscope, Switzerland (for Changins: e.g., Maltas et al., 2018; for Altwi,



Reckenholz: e.g., Hirte et al., 2021). All cropland sites were treated with mineral fertilizer according to the Swiss fertilization guidelines (Flisch et al., 2009) and regularly ploughed before sowing to a soil depth of approximately 25 cm. The managed peatlands were drained at the beginning of the 20th century and used as arable land for crop production. Crop types differ between each peatland site and season, with most of the cultivated crops being vegetables (Table S1).

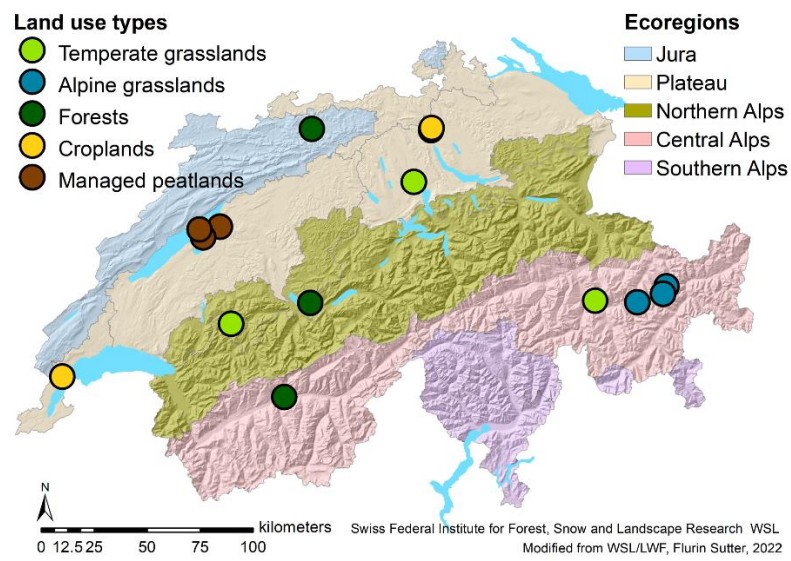

**Figure 1: Location of the study sites for each of the six dominant land-use types.**

### 2.2 In situ CO₂ flux measurements and gas sampling

We measured rates and sampled $CO_2$ from total soil respiration *in situ* at all sites during two sampling campaigns: one in summer 2023 (July – August) and another in late winter 2024 (March). These seasons were chosen to capture the greatest

variability in the soil environmental conditions. Winter measurements and sampling were conducted after snow melt. Severe climatic conditions precluded a sampling in March at the three alpine grassland sites.

At each of the 16 sites, we inserted three opaque cylindrical PVC frames (Ø = 30 cm; height = 15-25 cm; volume ~ 15 L) approximately 3-5 cm deep into the soil and 1-2 m apart to account for soil heterogeneity. Chambers were installed 1-2 weeks before sampling to allow soil respiration to re-equilibrate after installation disturbance. To exclude the contribution of above-

ground plant respiration, vegetation inside the chamber was clipped at installation and before sampling if plants had regrown. We measured $CO_2$ fluxes for each of the three chambers via a flow-through system using a LI-COR gas analyser (LI-8100A, LI-COR®). $CO_2$ fluxes were calculated as the rate of change in concentration over time in the chamber headspace using a linear approach while correcting for air pressure and mean chamber temperature using Gay-Lussac's ideal gas law (Butterbach-Bahl et al., 2011). Chamber closure times during flux measurements were 3-10 minutes. After flux measurements, chambers

were flushed with $CO_2$-free air five times the chamber volume (ca. 75 L) and sealed until $CO_2$ levels reached ~ 1000 ppm for subsequent [14]C analysis. Air from all three chambers was composited into a 2 L air bag (Cali-5-Bond, Calibrated Instruments,





LLC, USA). A detailed description of the sampling procedure is presented in Supplement S1 and Fig. S1. For $^{13}$C analysis, we sampled gas from each of the three chambers separately, transferring headspace air into pre-evacuated 12 mL Exetainer® vials using a 60 mL syringe. The lack of spatial and temporal replicates for each site is a consequence of the limited capacity of $^{14}$C

measurements. In winter, we repeated the $CO_2$ sampling and subsequent isotopic analysis for three spatial replicates at the forest sites Hölstein and Pfynwald. Results of the spatial replication are presented in Supplement S6, Fig. S8, and Table S7. For atmospheric background $^{14}CO_2$ measurements, ambient air (~ 20 cm above the soil surface) was sampled into 5 L air bags (Cali-5-Bond, Calibrated Instruments, LLC, USA). Soil environmental conditions were monitored before and after gas sampling. Ambient air temperature was measured at the height of the chamber (~ 15 cm above soil surface). Soil temperature

and volumetric water content (VWC) were monitored at 10 cm soil depth using a thermometer and HH2 moisture meter (Delta-T Devices Ltd), respectively.

**2.3 Soil sampling and analysis**

During the summer sampling campaign, we sampled soil at the centre of each PVC frame after the gas sampling was completed. Soil cores were taken with a Humax corer (Ø = 5 cm) down to 52 cm soil depth. Sampling was continued as deep as possible

(maximum depth = 90 cm) using a soil auger (Ø = 5 cm). Soil samples were stored at 2 °C until further processing. We split the mineral soil samples from the three soil cores into depth intervals of 0-5, 5-10, 10-20, 20-40, and 40-90 cm for all sites. At the forest sites, we additionally sampled organic layers (i.e., litter (L), fermented horizon (F), and humified horizon (H; only present at Beatenberg)). Alpine grassland sites were sampled separately by excavating soil profiles and taking volume-proportional samples for the pre-defined depth intervals. Volumes were estimated by simple measurements of the excavated

areas and samples were weighed and sieved to 4 mm in the field prior to processing in the laboratory.
For further analysis, we pooled and homogenized the soil of the three cores and corresponding depths to yield one composite sample per depth and site. In order to measure the released $CO_2$ by heterotrophic respiration during incubation, we sieved the fresh soil samples to 4 mm and removed roots and skeleton. After the incubations, we dried the samples at 40 °C and sieved them to 2 mm. Subsamples were ground to fine powder using a ball mill (MM2000, Retsch).

Gravimetric water content (GWC) was determined for each soil sample by drying a subsample of 6-10 g of fresh soil at 105 °C for 48 h. Bulk density was obtained by calculating the dry soil weight for each depth layer. Fine soil mass in each layer was determined by subtraction of the weight of the skeleton (> 2 mm). The volume of the skeleton was calculated by assuming a density of 2.65 g cm$^{-3}$ for stones (Walthert et al., 2002).

**2.4 Root and soil incubations**

We determined isotopic signatures of autotrophic and heterotrophic endmembers by conducting short-term root and soil incubations. Root incubations were prepared during the sampling campaigns *in situ*. We excavated fine roots (< 2 mm), including mycorrhizae within the chambers at 0-10 cm soil depth after the gas sampling. We carefully removed soil particles and rinsed the roots in an ultrapure water bath while keeping the root system as intact as possible. Roots from all three chambers




were combined into one composite sample and incubated instantly, as the $\delta^{13}$C value of excised roots can slightly change after
a few minutes (Midwood et al., 2006). We placed approximately 10 g of fresh roots into 2 L glass bottles that were wrapped in aluminium foil to exclude light. We immediately flushed the glass bottles with $CO_2$-free air until all ambient air was removed. Roots were incubated overnight at approximately 22 °C, followed by gas sampling into a 2 L air bag using a flow-through system. A detailed description of the sampling procedure from incubations is presented in Supplement S2. Gas for $^{13}$C analysis was sampled thereafter as described for *in situ* soil respiration.

Soil incubations were performed for each depth layer using the same procedure as for root incubations. Depending on the depth layer and their SOC contents, we incubated soil corresponding to dry soil weights between 30 and 250 g. Soils were incubated at 22 °C and field soil moisture levels. Depending on the respiration rates, the incubation time varied between one day and four weeks. We calculated basal respiration rates for each soil incubation by integrating the entire incubation period. Weighted respiration rates were then calculated for each depth layer by accounting for dry mass in the respective depth layer
and adjusted to field temperature assuming a Q10 of 2.4 (Raich & Schlesinger, 1992), a temperature decrease with soil depth of 0.035 °C cm$^{-1}$ in summer, and a temperature increase of 0.036 °C cm$^{-1}$ in winter (Bourletsikas et al., 2023). Basal respiration was used as a proxy for SOC decomposability by relating it to the SOC content.

## 2.5 Isotopic analysis of gas samples

The $^{13}CO_2$ contents of all gas samples were measured by an isotope-ratio mass spectrometer (IRMS Gas-Bench II coupled with
a Delta-V Advanced IRMS, Thermo GmbH, Germany) at the Swiss Federal Research Institute of Forest, Snow and Landscape WSL. Graphitization and $^{14}$C measurements were performed at the Laboratory of Ion Beam Physics, ETH Zurich, Switzerland. For $^{14}$C analysis, gas samples were graphitized using an Air Loading Facility (ALF; Gautschi, 2017) coupled to an Automated Graphitization Equipment (AGE3, ETH Zurich, Switzerland; Wacker, Němec, et al., 2010) with an integrated zeolite trap to adsorb $CO_2$ from the sampling bag. The $^{14}CO_2$ contents of all gas samples were measured using a MIni radioCArbon DAting
System (MICADAS, ETH Zurich, Switzerland; Synal et al., 2007) or a Low Energy AMS (LEA, ETH Zurich & IonPlus AG, Switzerland; Ramsperger et al., 2024). Measurement uncertainties were < 2 ‰. For data evaluation, the standard Oxalic Acid II (Mann, 1983) and blank material from the $^{14}$C-free phthalic anhydride (PhA) were measured alongside the samples and evaluated with the BATS software (Wacker, Christl, et al., 2010).

## 2.6 Isotopic analysis of bulk soil

Bulk soil samples were analysed for total and organic C, and $\delta^{13}$C by dry combustion with an automated elemental analyser – continuous flow isotope ratio mass spectrometer (Euro-EA 3000, HEKAtech GmbH, Germany, interfaced with a Delta-V Advanced IRMS, Thermo GmbH, Germany). Measurements were corrected against primary reference materials (VPDB and AIR). Measurement uncertainties were < 0.3 ‰. Samples with $\delta^{13}$C values exceeding -25 ‰, which suggest a potential contribution of inorganic C, were additionally analysed after fumigation with 37 % HCl to remove inorganic C (Walthert et
al., 2010).



For [14]C analysis, potential inorganic C was removed for all samples by fumigation with 37 % HCl (Komada et al., 2008; Walthert et al., 2010). Samples were acidified for 72 h at 60 °C and neutralized with NaOH pellets (72 h, 60 °C). All glassware was combusted at 550 °C for 5 h prior to use. [14]C measurements in SOC of bulk soil were performed on a MICADAS (ETH Zurich, Switzerland; Synal et al., 2007) featuring a gas ion source and coupled to an Elemental Analyser (EA vario MICRO

cube, Elementar, Germany; Ruff et al., 2010) at the Laboratory of Ion Beam Physics, ETH Zurich, Switzerland. Measurement uncertainties were 6-8 ‰.

Comparisons of isotopic signatures from heterotrophically respired $CO_2$ and SOC revealed possible contributions of carbonate weathering to $CO_2$ for some sites and depths. We assumed contributions of carbonate weathering in cases where $\Delta^{14}C$ signatures indicated $CO_2$ age > SOC age and/or in cases where $\delta^{13}CO_2 > \delta^{13}C$ of SOC (except croplands and managed

peatlands). Although we estimated carbonate contributions with certain limitations (Supplement S7), we were not able to accurately correct isotopic signatures of $CO_2$ from heterotrophic respiration because of limited information on endmember isotopic signatures (i.e., carbonate, SOC-derived $CO_2$). Nevertheless, we provide estimated contributions and corrected $\Delta^{14}CO_2$ signatures in Table S3. Generally, our estimations revealed that the mean effect of carbonate weathering on total heterotrophic respiration and on *in situ* soil respiration was < 5±3 % and < 2±1 %, respectively (Table S3).

**2.7 Estimation of ages from $\Delta^{14}CO_2$ signatures**

We estimated ages of $CO_2$ from heterotrophic respiration (total, topsoil and subsoil layers) and total soil respiration using carbonate-corrected $\Delta^{14}CO_2$ values with the web-based OxCal software (2024; Bronk Ramsey, 2009) using the Bomb21NH1 calibration curve (Hua et al., 2022). In cases where the $\Delta^{14}CO_2$ values were outside the calibration range (younger than 2019), ages were derived by comparing $\Delta^{14}CO_2$ values with extended bomb curve data. The curve Bomb21NH1 was extended with

$\Delta^{14}CO_2$ values measured at Jungfraujoch, Switzerland by ICOS between 2022 to 2023 (Emmenegger et al., 2024) and in 2024 by Geissbühler et al., in prep.. A detailed description of the approach is presented in Supplement S3.

**2.8 Source partitioning using Bayesian mixing models and correction for atmospheric background**

The soil respired $CO_2$ (SR) that we captured *in situ* from the chamber headspace was partitioned into autotrophic (AR) and heterotrophic (HR from each depth layer) sources, as well as residual atmospheric air (ATM) using Bayesian mixing model

approaches: MixSIAR (R package MixSIAR, version 3.1.12; Moore & Semmens, 2008; Stock et al., 2018) and an implementation using Python. Partitioning was performed for each site separately. The models use Markov Chain Monte Carlo to sample possible distributions so that:

$$\Delta^{14}CO_{2_{SR}} = f_{ATM}\Delta^{14}CO_{2_{ATM}} + f_{AR}\Delta^{14}CO_{2_{AR}} + \sum_{depth=1}^{n} f_{HR,depth}\Delta^{14}CO_{2_{HR,depth}} \tag{1}$$



$$\delta^{13}CO_{2_{SR}} = f_{ATM}\delta^{13}CO_{2_{ATM}} + f_{AR}\delta^{13}CO_{2_{AR}} + \sum_{depth=1}^{n} f_{HR,depth}\delta^{13}CO_{2_{HR,depth}} \tag{2}$$

$$1 = f_{ATM} + f_{AR} + \sum_{depth=1}^{n} f_{HR,depth} \tag{3}$$

where $0 \leq f \leq 1$ is the proportional contribution of each endmember to total soil respiration and $n$ is the number of incubated depth layers. Input data were $\Delta^{14}CO_2$ values and associated measurement errors ($\sim 2$ ‰) as well as $\delta^{13}CO_2$ values and associated standard deviations of three replicates ($\sim 0$–$1$ ‰) for mixtures and sources. Constraints and modification of input data are presented in Supplement S4. In the Bayesian mixing model in Python, we treated heterotrophic respiration from each depth layer as a separate source and added weighted respiration rates as deterministic predictors (assumed uncertainty = 10 %). In MixSIAR, we aggregated depths to topsoil (0-5, 5-10 cm) and subsoil (10-20, 20-40, 40-maximum depth cm) layers to reduce the number of sources in favour of the model performance. Respective isotopic endmember values (topsoil and subsoil) were calculated using a mass balance approach. Isospace plots of MixSIAR are presented in Fig. S9-Fig. S17. In MixSIAR we used weighted contributions of topsoil and subsoil to total heterotrophic respiration to set up an informative prior with the flat Dirichlet distribution, so that: $\alpha = (f_{ATM}, f_{AR}, \sum f_{HR,depth})$ corresponds to $\alpha = (1, 1, 1)$. Model convergence was assessed via Gelman-Rubin (values < 1.05) and Geweke diagnostics. Further information on model settings is given in Table S2. We combined the mean values and standard deviations of both models to account for specific model restrictions and variability in model outputs. A detailed description of the differences and limitations of the two approaches (MixSIAR and implementation in Python) is given in Supplement S5.

To exclude atmospheric contribution to total soil respiration, we adjusted mean values and standard deviations of autotrophic and heterotrophic contributions so that:

$$f_{AR_{corr}} = \frac{f_{AR}}{1 - f_{ATM}} \tag{4}$$

$$\sum_{depth=1}^{n} f_{HR,depth_{corr}} = \frac{\sum_{depth=1}^{n} f_{HR,depth}}{1 - f_{ATM}} \tag{5}$$

We further corrected $\Delta^{14}CO_2$ and $\delta^{13}CO_2$ values of *in situ* soil respiration for the effect of atmospheric background using the proportions of source contributions from the model outputs. We calculated the corrected isotopic values ($SR_{corr}$) as:

$$\Delta^{14}CO_{2_{SR_{corr}}} = \frac{\Delta^{14}CO_{2_{SR}} - (\Delta^{14}CO_{2_{ATM}} \times (1 - f_{SR}))}{f_{SR}} \tag{6}$$

$$\delta^{13}CO_{2_{SR_{corr}}} = \frac{\delta^{13}CO_{2_{SR}} - (\delta^{13}CO_{2_{ATM}} \times (1 - f_{SR}))}{f_{SR}} \tag{7}$$

where $f_{SR}$ is the fraction of soil-derived $CO_2$ calculated as the sum of autotrophic and heterotrophic source proportions as initially derived from the model outputs.





# 3 Results

## 3.1 *In situ* soil respiration rates across land-use types and seasons

Average *in situ* respiration rates across land-use types ranged from 125 to 363 mg $CO_2$-C $m^{-2}$ $h^{-1}$ in summer and from 91 to 191 mg $CO_2$-C $m^{-2}$ $h^{-1}$ in winter (Table 1, Fig. 2). Forests exhibited the lowest respiration rates in both seasons. In summer, managed peatlands had the highest respiration rates, however, this was mainly related to very high respiration at one of the sites (Lindergut). While the effects of land-use types on *in situ* soil respiration were not significant ($p = 0.133$), seasonality had a significant effect ($p < 0.001$) (Table S5). In agreement with the seasonality, *in situ* respiration rates were affected

significantly by soil temperature ($p < 0.001$) and soil water content ($p = 0.002$) (Table S4). While grasslands had soil water contents above 45 vol-%, forest and cropland sites exhibited particularly low soil water contents in summer (< 20 vol-%; Fig. 1).

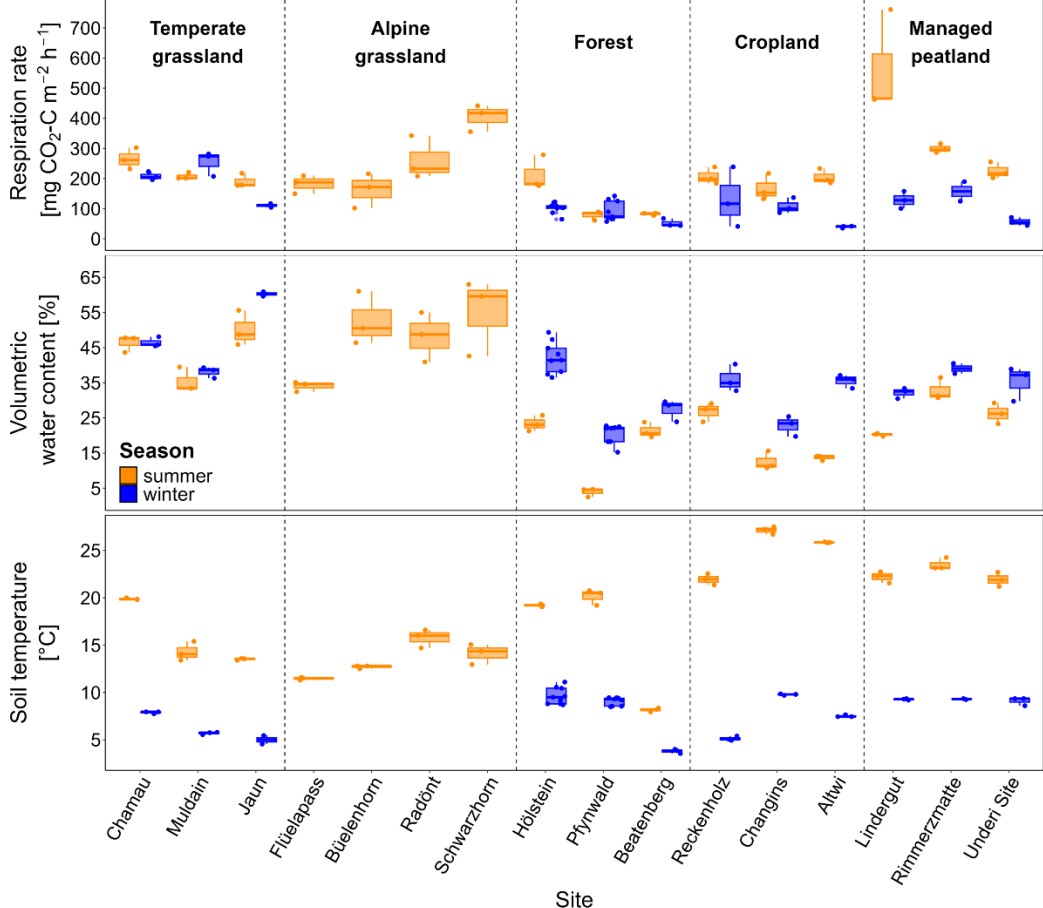

**Figure 2: Soil respiration rates (mg $CO_2$-C $m^{-2}$ $h^{-1}$), volumetric water content (VWC; vol-%), and soil temperature (°C) from in situ**
**measurements in summer (orange) and winter (blue). VWC and soil temperature are presented for 0–10 cm of the mineral soil (excluding organic layers in forests).**




### 3.2 Δ $^{14}$CO$_2$ values in total soil respiration across land-use types and seasons

The soil respired $\Delta^{14}CO_2$ values measured *in situ* varied significantly across land-use types ($p = 0.001$ and $p = 0.002$ with and without managed peatlands (Table S5), with the highest $\Delta^{14}CO_2$ values in forests, followed by grasslands, croplands, and

managed peatlands. The latter showed strongly $^{14}$C-depleted values (i.e., $\Delta^{14}CO_2 < -50$ ‰; Fig. 3).

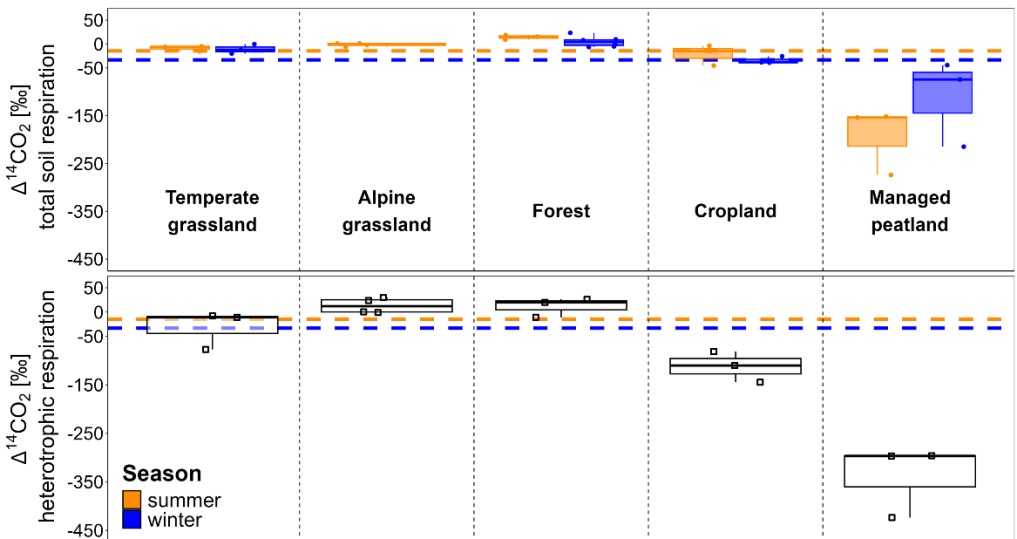

**Figure 3: Δ$^{14}$CO$_2$ values (‰) of total soil respiration (top) in summer (orange) and winter (blue) and heterotrophic respiration (bottom) across land-use types. Dashed horizontal lines indicate mean atmospheric background Δ$^{14}$CO$_2$ levels across land-use types in summer (orange) and winter (blue). Note that Δ$^{14}$CO$_2$ values of some sites and depths might be**
**affected by carbonate weathering, lowering these values. However, the overall effect on total heterotrophic and total soil respiration is small and trends remain the same (Table S3).**

The *in situ* soil-respired $^{14}$CO$_2$ exhibited modern levels in croplands and grasslands, corresponding to ages < 10 years (Table 1). Within grasslands, $\Delta^{14}CO_2$ values significantly increased ($p = 0.006$) with elevation (Fig. S2, Table S6). Forest soils released CO$_2$ with positive $\Delta^{14}$C values, indicative of bomb-derived, decadal-old CO$_2$ (Table 1). In managed peatlands, soil-respired
CO$_2$ had very low $\Delta^{14}CO_2$ values, corresponding to mean ages of approximately 1500 years (Table 1).

The $\Delta^{14}CO_2$ values of total soil respiration differed significantly between summer and winter ($p = 0.001$) with $\Delta^{14}CO_2$ values being generally closer to atmospheric CO$_2$ in winter. However, the effect of seasonality varied significantly across land-use types ($p < 0.001$). While managed peatlands exhibited the largest seasonal changes in soil respired $\Delta^{14}CO_2$, temperate grasslands and forests showed the lowest seasonality (Fig. 3).




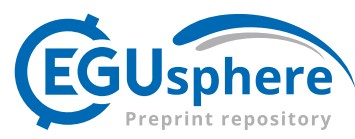

**Table 1: Mean soil respiration rates, $\Delta^{14}CO_2$ values, estimated ages and relative source contribution of heterotrophic and autotrophic sources. Carbonate-corrected $\Delta^{14}CO_2$ values were used for the estimation of ages (Table S3).**

| | Temperate grassland | | Alpine grassland | | Forest | | Cropland | | Managed peatland | |
|---|---|---|---|---|---|---|---|---|---|---|
| **Season** | summer | winter | summer | winter | summer | winter | summer | winter | summer | winter |
| **Total soil respiration** | | | | | | | | | | |
| Respiration rate [mg $CO_2$-C m$^{-2}$ h$^{-1}$] | 221 ± 39 | 191 ± 73 | 253 ± 110 | - | 125 ± 76 | 91 ± 29 | 193 ± 22 | 93 ± 48 | 363 ± 178 | 144 ± 52 |
| $\Delta^{14}CO_2$ value [‰] | -9 ± 6 | -11 ± 10 | -1 ± 3 | - | 14 ± 5 | 5 ± 10 | -21 ± 21 | -34 ± 7 | -193 ± 70 | -111 ± 91 |
| Estimated age [yrs] | 8 ± 1 | 7 ± 1 | 5 | - | 11 ± 1 | 9 ± 1 | 162 ± 12 / 7 ± 1* | 230 ± 32 / 5* | 1486 ± 47 | 828 ± 23 |
| **Heterotrophic respiration** | | | | | | | | | | |
| Estimated $\Delta^{14}CO_2$ value [‰] | | | | | | | | | | |
| Total | -32 ± 39 | | 13 ± 16 | | 12 ± 20 | | -112 ± 32 | | -339 ± 73 | |
| Organic layer | | | | | 6 ± 18 | | | | | |
| Topsoil | 1 ± 13 | | 31 ± 22 | | 23 ± 16 | | -109 ± 62 | | -167 ± 27 | |
| Subsoil | -56 ± 58 | | -41 ± 17 | | -54 ± 83 | | -107 ± 50 | | -349 ± 75 | |
| Estimated age [yrs] | | | | | | | | | | |
| Total | 9 ± 1 | | 40 ± 3 | | 11 ± 1 | | 648 ± 32 | | 3188 ± 46 | |
| Organic layer | | | | | 7 | | | | | |
| Topsoil | 7 | | 12 ± 1 | | 11 ± 1 | | 241 ± 16 | | 1394 ± 26 | |
| Subsoil | 39 | | 387 ± 14 | | 142 ± 4 | | 729 ± 19 | | 3294 ± 48 | |
| Rel. source contribution [%] | | | | | | | | | | |
| Total | 40 ± 5 | 46 ± 5 | 37 ± 5 | - | 69 ± 7 | 53 ± 4 | 31 ± 1 | 11 ± 2 | 53 ± 7 | 23 ± 4 |
| Organic layer | | | | | 21 ± 4 | 34 ± 2 | | | | |
| Topsoil | 25 ± 4 | 21 ± 2 | 29 ± 4 | - | 44 ± 5 | 5 ± 0 | 18 ± 1 | 3 ± 2 | 5 ± 5 | 3 ± 3 |
| Subsoil | 15 ± 2 | 24 ± 4 | 9 ± 2 | - | 5 ± 0 | 16 ± 0 | 12 ± 1 | 8 ± 1 | 48 ± 4 | 20 ± 2 |
| **Autotrophic respiration** | | | | | | | | | | |
| Estimated $\Delta^{14}CO_2$ value [‰] | -12 ± 4 | -15 ± 6 | -6 ± 3 | - | -4 ± 4 | -9 ± 5 | -13 ± 3 | -32 ± 4 | -14 ± 4 | -34 ± 3 |
| Rel. source contribution [%] | 60 ± 7 | 54 ± 5 | 63 ± 6 | - | 31 ± 5 | 47 ± 3 | 69 ± 6 | 89 ± 3 | 47 ± 6 | 77 ± 3 |

*For total soil respiration in croplands, two age ranges are possible. Highly $C^{14}$ depleted atmospheric background samples for these sites make it difficult to interpret the depleted $\Delta^{14}CO_2$ values of total soil respiration. The values are within a range that the comparison with the site-adjusted bomb curves would reveal ages of around 7 and 5 years in summer and winter, respectively. The low relative source proportion of heterotrophic respiration is further indicative of an overall respiration of younger $CO_2$.





### 3.3 Isotopic signatures of atmosphere, autotrophic, and heterotrophic respiration

Our results showed seasonal variation in $\Delta^{14}CO_2$ values of atmosphere and autotrophic respiration (Fig. S3). In the atmosphere,
the mean $\Delta^{14}CO_2$ value was more depleted in winter than in summer at all sites (mean across all sites: -14±10 ‰ in summer and -33±10 ‰ in winter). Autotrophic $\Delta^{14}CO_2$ signatures were generally close to atmospheric $\Delta^{14}CO_2$ values and followed their seasonal trend with more depleted $\Delta^{14}CO_2$ values in winter. In forests, however, autotrophic $\Delta^{14}CO_2$ signatures in winter were similar to those in summer (Fig. S3).

Generally, the $\Delta^{14}CO_2$ values from heterotrophic respiration indicate a continuous increase in the age of respired $CO_2$ with
increasing soil depth, which was consistent across all land-use types (Fig. 4). However, $\Delta^{14}CO_2$ values in croplands and managed peatlands increased from very depleted values at 0-5 cm to less depleted values at 5-10 cm soil depth.

The variation of $\Delta^{14}CO_2$ values of total heterotrophic respiration across land-use types followed a similar pattern as that of total soil respiration. Temperate grassland soils released the most contemporary $CO_2$ with ages < 10 years, while alpine grassland and forest soils released bomb-derived, decadal-old $CO_2$ with average ages of ~ 40 and 11 years, respectively (Table
1). Within grasslands, heterotrophic respiration from topsoil layers exhibited an even stronger increase of $\Delta^{14}CO_2$ values with elevation than *in situ* soil respired $CO_2$ ($p = 0.056$; Fig. 4, Fig. S2, Table S6). In contrast, heterotrophic $CO_2$ in subsoil layers did not show an elevational pattern. $CO_2$ released from subsoil had depleted $\Delta^{14}C$ values for all grassland sites (Fig. 4). Cropland soils released ~ 650-year-old $CO_2$ and managed peatland even emitted ~ 3200-year-old $CO_2$ ($p_{land\text{-}use\ type} < 0.001$; Fig. 3, Table 1, Table S5).





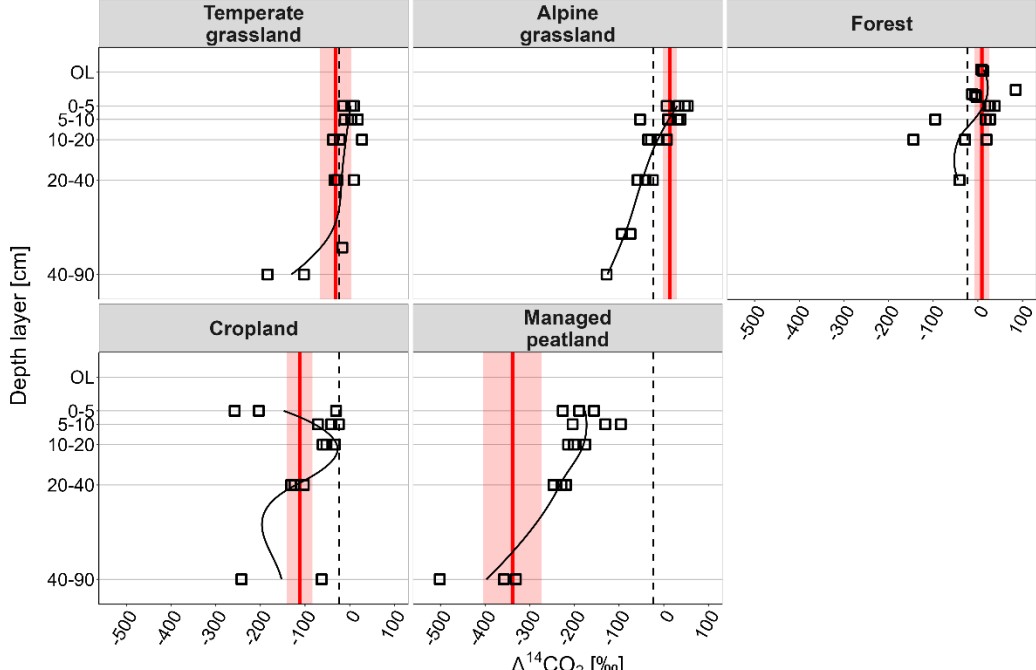

**Figure 4: Depth profiles of $\Delta^{14}CO_2$ signatures (‰) of heterotrophic respiration (squares) and weighted $\Delta^{14}CO_2$ values of total heterotrophic respiration (red line) and error range (red shaded area). Dashed black vertical lines indicate the mean atmospheric $\Delta^{14}CO_2$ value during our sampling campaigns across all land-use types and both seasons (-24 ‰).**

## 3.4 SOC decomposability

SOC decomposability, measured by the $CO_2$ release from soils under standardized conditions, varied significantly across land-use types in both topsoil (p < 0.001) and subsoil (p = 0.002), following a general pattern of decreasing decomposability with increasing soil depth (Fig. 5, Table S5). Croplands and managed peatlands exhibited the lowest values (< 20–60 µg $CO_2$-C $g^{-1}$ SOC $h^{-1}$), while forests showed high decomposability in the organic layer (> 200 µg $CO_2$-C $g^{-1}$ SOC $h^{-1}$), but much lower levels in the mineral soil. Temperate grasslands had intermediate decomposability in topsoil and lower values in subsoil. Alpine grasslands exhibited the highest values (46–257 µg $CO_2$-C $g^{-1}$ SOC $h^{-1}$), with increased decomposability at higher elevations.



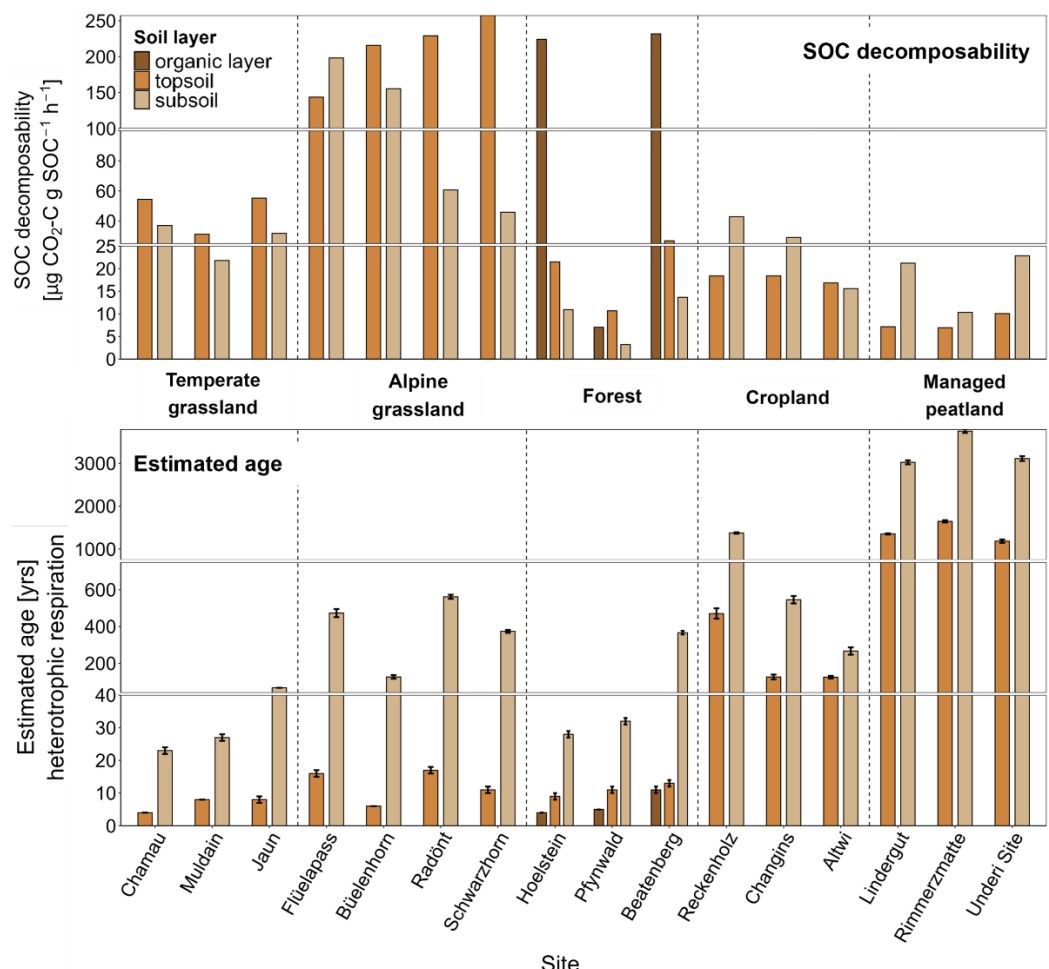

**Figure 5: SOC decomposability (top; µg CO$_2$-C g SOC$^{-1}$ h$^{-1}$) and estimated ages of heterotrophic respiration (bottom) across land-use types for organic layers (dark brown), topsoil (red brown), subsoil (beige). Note that the breaks in the y-axes have different scales for better visualisation.**

### 3.5 Changes in source contribution across land-use types

The $\Delta^{14}CO_2$ signatures of total soil respiration fell between $\Delta^{14}CO_2$ values of autotrophic and heterotrophic endmembers, indicating both contributed to total soil respiration (Fig. S5). Although the difference in source contribution across land-use types was not statistically significant ($p = 0.071$), distinct patterns existed (Fig. 6, Table S5). Overall, autotrophic respiration was the dominant source of total soil respiration across all land-use types and both seasons (~ 60 %) (Fig. 6, Table 1). The average autotrophic contribution was greatest in croplands (~ 80 %) followed by managed peatlands and grasslands (~ 60 %). Forest showed the smallest average autotrophic contribution (~ 40 %) and accordingly, was the only land-use type where heterotrophic respiration dominated total soil respiration. The relative contribution of heterotrophic respiration was generally higher in summer than in winter for all land-use types except croplands (Fig. 6, Table 1). Managed peatlands showed the





strongest seasonal variation of source contribution with an average heterotrophic contribution of 47 % in summer and 17 % in

winter.

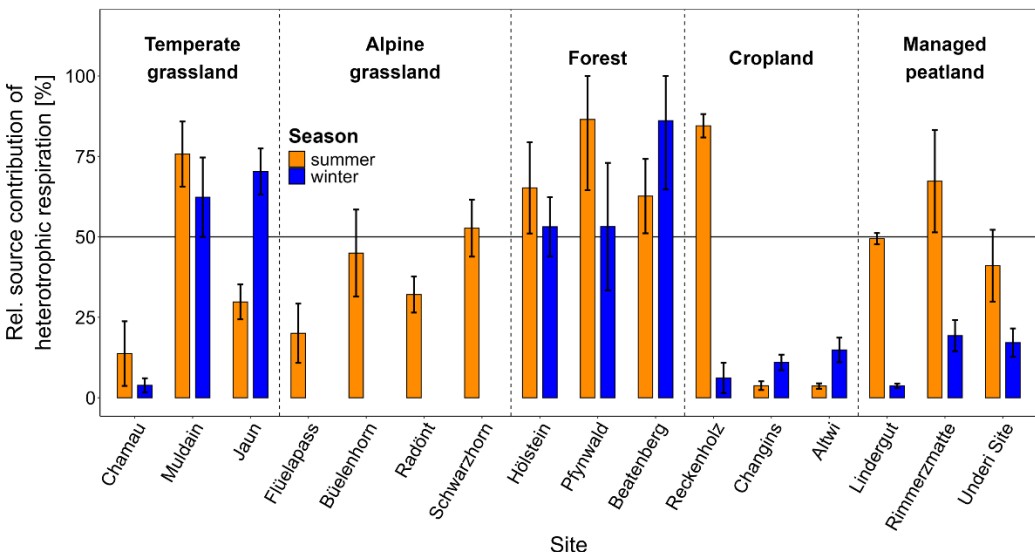

**Figure 6: Relative source contribution of heterotrophic respiration in summer (orange) and winter (blue) across all land-use types and individual sites. Bars represent combined mean values and error bars combined standard deviations from the model outputs.**

Measurements of the isotopic composition of heterotrophically respired $CO_2$ across various depths revealed differences in the

contributions of organic layer, topsoil, and subsoil layers to total soil respiration among land-use types. Topsoils were the

dominant heterotrophic source in grasslands (~ 26 %) in both seasons and in forests during summer (44 %) (Table 1, Fig. 4).

In forests, the organic layer contributed substantially to total soil respiration in both seasons (21 % in summer and 34 % in

winter). In croplands, the generally low contribution of heterotrophic respiration showed similar contributions from top- and

subsoil layers (Table 1, Fig. 4). Managed peatlands were almost entirely dominated by heterotrophic respiration from subsoil

layers in summer (48 % of total soil respiration) while subsoil contributions were significantly reduced in winter (Table 1).

## 4 Discussion

Our $\Delta^{14}CO_2$ measurements clearly document that *in situ* soil respiration from all land-use types, except from managed

peatlands, was dominated by relatively young C (< 12 years old). In addition, mineralized $CO_2$ from total heterotrophic

respiration showed young C release (< 40 years old) from all land use types, except from managed peatlands and croplands.

This pattern indicates that when C enters the soil system, the largest fraction gets rapidly respired back to the atmosphere.

Although soil-respired $CO_2$ was predominantly young, estimated mean ages varied from < 10 years in grasslands to ~ 10 years

in forests, and reached up to ~ 1500 years in managed peatlands. In conjunction with variations in soil respiration rates and

SOC decomposability across land-use types, the $^{14}$C-derived ages in soil-respired $CO_2$ provide evidence that the magnitude,

velocity, and pathways of C cycling within the plant-soil regime vary among these systems. Although there was variation in



environmental conditions (i.e. soil moisture, soil temperature) within individual land-use types across different sites, the overarching patterns distinguishing land-use types remained consistent and site-level variability was accounted for in the statistical analysis. This suggests that land use played a more significant role in shaping respiration dynamics than site-specific environmental conditions.

Here, we propose that SOC decomposability, *in situ* soil respiration rates and $^{14}$C-derived ages of soil-respired $CO_2$ can serve as comprehensive indicators to categorize C cycling into five different systems: (1) high-throughput systems characterized by rapid C cycling – temperate grasslands, (2) retarding systems characterized by slow C cycling – alpine grasslands, (3) preserving systems with a delayed transfer of assimilates back to the atmosphere through soil respiration – forests, (4) destabilized C-depleted systems, where reduced C inputs result in the depletion of recent C material and SOC stocks –

croplands, and (5) destabilized hotspots that release old stored soil C due to significant disturbances in natural C cycling – managed peatlands. C stabilization and retention are greater in natural ecosystems (forests, alpine grasslands), whereas disturbances due to agricultural management, land-use change (especially croplands, managed peatlands) and climate change (as reflected in differences between temperate and alpine grasslands) lead to C destabilization or a decline in ecosystem retention capacity (e.g., alpine grasslands under climate warming) (Fig. 7).

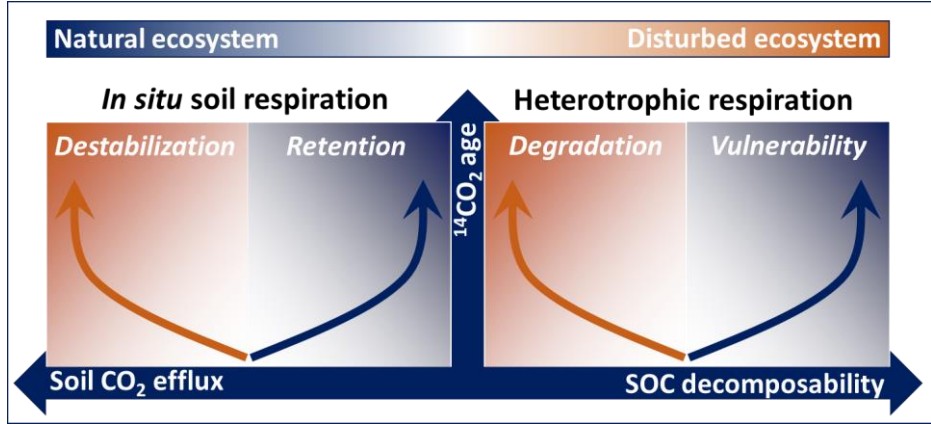


**Figure 7: Conceptual framework to categorize the status of C cycling in land-use systems as a function of the rate and age of soil-respired $CO_2$.**

## 4.1 High-throughput systems – temperate grasslands

Among all land-use types, temperate grasslands represent the land-use type with the highest rates and the youngest age of soil-

respired $CO_2$, characterizing them as high-throughput systems. The high respiration rates in grasslands are consistent with those observed across European land-use types (Schaufler et al., 2010). Here, we show that temperate grasslands have high contributions of autotrophic respiration (60 %), indicating that a large fraction of their high GPP (Schulze et al., 2009) is rapidly allocated to the belowground, where it is metabolized and respired. Among land-use types, grasslands typically show the densest rooting system, the highest root turnover (E. F. Solly et al., 2014), rapid belowground C allocation (Fuchslueger et

al., 2016) and a particularly high rhizodeposition (R. Wang et al., 2021). Our study reveals young ages of heterotrophically



respired $CO_2$ and in bulk SOC in topsoil layers (Fig. 4, Fig. S6), indicating that plant-derived C inputs into the soil are rapidly processed by soil microbes. Overall, this results in rapid C cycling through the plant-soil system in temperate grasslands.

## 4.2 Retarding systems – alpine grasslands

Our assessment of grassland sites, spanning an elevation gradient from 390 to 2630 m a.s.l., shows increasing ages of soil-
respired $CO_2$ and SOC with elevation (Fig. S2). Since autotrophic respiration dominated irrespective of elevation, the increasing age of *in situ* soil-respired $CO_2$ with elevation derives from microbial processing of older SOC at higher elevation. The $\Delta^{14}CO_2$ values of heterotrophic respiration in topsoil layers increased with elevation, indicating enhanced respiration of bomb-derived C as well as stronger relative contributions of topsoil layers with increasing elevation. This provides evidence of a slowing of C cycling towards colder climatic conditions and lower pH in alpine grasslands with shallower soils (Table
S1) (Chen et al., 2021; Leifeld et al., 2009). In support, (Leifeld et al., 2009) observed an increasing age in particulate organic matter in grassland soils especially in deeper soils, which was related to the reduced productivity in alpine grasslands and shallower rooting depth at low mean annual temperatures. Our finding of the highest SOC decomposability in alpine grasslands among all land-use systems shows that these soils contain the largest amount of readily decomposable SOC (Fig. 5). Along with their high $^{14}$C ages in SOC, this suggests that the retarded processing of C inputs into the soil has led to the retention and
accumulation of labile C. This, in turn, indicates that alpine grasslands are particularly vulnerable to C cycle perturbations induced by climate warming or by disturbances, potentially leading to losses of old but non-stabilized C.

## 4.3 Preserving systems - forests

In comparison to the other land-use types, forests showed the lowest respiration rates, the greatest contribution of heterotrophic respiration and released the oldest $CO_2$ via *in situ* soil respiration among undisturbed ecosystems. A probable reason for the
low respiration rates and the retarded assimilate transfer to the belowground is the allocation of a larger proportion to aboveground biomass (Schaufler et al., 2010) and the retarded transfer to the rhizosphere (Gao et al., 2021). Beyond the lower and slower belowground allocation, elevated $\Delta^{14}CO_2$ values of heterotrophic respiration indicate that the release of older $CO_2$ from forest soils also derives from the decomposition of decadal-old SOC compared with temperate grasslands (Fig. S5). This, in turn, implies slower turnover of plant-derived C in SOC in forests as compared to other land-use types (for croplands and
managed peatlands the very old age of heterotrophically respired $CO_2$ primarily derives from the decomposition of preserved, old C). While slower turnover in alpine grasslands can be associated with colder climatic conditions, higher ages of SOC and respired $CO_2$ in forests likely derive from both the input of older below- and aboveground C to the soil (E. Solly et al., 2013) and its lower quality slowing decomposition. Forest litter is often enriched in lignin and polyphenols, and nutrient-poorer than grassland litter, and thus more recalcitrant (Zhang et al., 2008). In addition, an acidic soil environment induced by needle litter
in coniferous forests together with cold and moist conditions, suppresses microbial and faunal activity (Desie et al., 2020; Zhang et al., 2008), which results in a build-up of thick organic layers (as in Beatenberg) (Hiltbrunner et al., 2013). Our source partitioning revealed that organic layers were major contributors to total heterotrophic respiration (~ 30 % in summer and ~



60 % in winter). Combined with the substantial contributions of total heterotrophic respiration (53-69 %), this resulted in the highest $^{14}$C-derived age of *in situ* soil-respired $CO_2$ among semi-natural ecosystems.

## 4.4 Destabilized C-depleted systems – croplands

Our results show that while croplands had $\Delta^{14}CO_2$ values in *in situ* soil respiration close to atmospheric $\Delta^{14}CO_2$ values, their heterotrophically respired $CO_2$ and bulk SOC had pre-bomb $^{14}$C levels and were thus depleted in $^{14}$C compared to forests and grasslands (Fig. 3, Fig. S6). This overall pattern reveals a high autotrophic contribution in croplands, releasing contemporary C, as well as the long-term depletion of SOC. The latter is likely caused by continuous biomass removal during harvest and the small belowground C input reducing SOC storage (Don et al., 2011; Keel et al., 2019). Losses of SOC can additionally be fostered by soil tillage destroying soil aggregates (e.g., Six et al., 1999), generally leaving SOM with a relatively larger fraction of persistent, old C. In agreement, our study shows that along with the depletion of $^{14}$C in soil respired $CO_2$ and SOC compared to forests and grasslands, cropland soils also showed a low SOC decomposability and the smallest SOC stocks among all land-use types (Fig. 6, Fig. S7). This suggests that these cropland soils are destabilized, aligning with observations from Swiss long-term experiments that show declining SOC stocks in croplands, which had previously been used as grasslands (Keel et al., 2019). It should be noted that the $CO_2$ released from two cropland sites (Altwi, Changins) is likely affected by carbonate weathering induced by liming practices, which led to relatively high $^{13}CO_2$ and highly depleted $^{14}CO_2$ values. However, isotope mixing revealed that potential carbonate weathering contributed < 1 % to total soil respiration in croplands. This is consistent with e.g., (Schindlbacher et al., 2015, 2019; Serrano-Ortiz et al., 2010) who suggest that $CO_2$ release from carbonates makes a minor contribution to soil-respired $CO_2$. In addition, potential carbonate contribution did not affect estimates of autotrophic contributions to total soil respiration. $^{14}$C-derived ages were estimated using carbonate-corrected $^{14}$C values (Supplement S6, Table S3).

Autotrophic contributions in croplands found in this study (summer: 69 %, winter: 89 %) align with previous studies using $^{14}$C and $^{13}$C labelling approaches (Søe et al., 2004; Werth & Kuzyakov, 2008). Low heterotrophic contributions are likely due to small litter inputs and poor decomposability of the remaining old SOC (Fig. S6). We further assume that high relative autotrophic contributions are also related to hampered heterotrophic respiration due to adverse soil environmental conditions. The two sites, Altwi and Changins, with exceptionally high autotrophic contributions (96 %), exhibited very low water content (< 15 vol-%). In contrast, the Reckenholz site, with moderate moisture levels (27 vol-%), showed a much lower autotrophic contribution of only 15 % (Fig. 2, Fig. 6). The low soil water contents in croplands are probably due to the sparse vegetation cover which enhances evaporative water losses.

## 4.5 Destabilized hotspots of C release – managed peatlands

The high release of millennial-old $CO_2$ identified drained, managed peatlands as hotspots for losses of old C release despite the dominant contribution of autotrophic respiration to total soil respiration (mean across all managed peatland sites: 68±19 %). The release of old, pre-bomb $CO_2$ has also been observed for other drained and managed peatlands in Switzerland (Bader

 

et al., 2017; Y. Wang et al., 2021) and can be related to aeration following drainage, which induces decomposition of peat material that had previously been protected by anaerobic water-saturated conditions. At our sites, the low SOC decomposability in conjunction with high $^{14}$C-derived ages (Fig. 6) indicates an advanced degradation stage of the destabilized peat soil.

The autotrophic contribution (47 % in summer, 77 % in winter) to total soil respiration that we estimated for managed peatlands agrees well with contributions found in natural peatlands during the growing season, where $^{14}$C pulse-labelling and trenching

approaches revealed autotrophic respiration to contribute between 35 and 61 % (Crow & Wieder, 2005; Wunderlich & Borken, 2012) to total soil respiration. The observed seasonal differences in autotrophic contributions support that water table depth plays a crucial role for the sources of respired $CO_2$ (Rankin et al., 2023; Stuart et al., 2023). Counterintuitively, the autotrophic contribution to soil respiration was lower in summer than in winter, despite the high plant activity during the growing season. Lower heterotrophic contributions in winter (Table 1) are likely because of higher water tables (Paul et al., 2021, 2024). Drier

conditions in summer aerate larger portions of the peat, thereby enhancing peat decomposition (Rankin et al., 2023; Stuart et al., 2023). Source partitioning further revealed higher heterotrophic contribution from subsoil layers than from topsoil layers in managed peatlands, which contrasts with observations from all other land-use types (Table 1). Higher subsoil contributions likely result from pronounced increases in SOC stocks with depth (Fig. S10, Fig. 6). Respiratory C losses from deeper peat also contribute to the high *in situ* $^{14}CO_2$-ages of ~ 1500 years released from the managed peatlands in summer.

**4.6 Seasonal dynamics in the age of soil respired $CO_2$ across land-use types**

In winter, $\Delta^{14}CO_2$ values were generally closer to atmospheric levels than in summer which we attribute to higher contributions of autotrophic respiration in winter. Our results are in contrast to findings from other studies that found higher contributions of heterotrophic respiration in winter related to winter dormancy of plants (Dörr & Münnich, 1986; Schindlbacher et al., 2009; Torn et al., 2009; Wang et al., 2000). However, since autotrophic respiration is rather driven by plant phenology than by

temperature (Atarashi-Andoh et al., 2012), we likely captured an active state of the phenology in March for grassland and forest sites. In addition, (Marchand et al., 2025) revealed that root activity of deciduous tree species is not constrained by low temperatures in winter. In forests, evergreen vegetation, as well as mosses and graminoids on the forest floor might have additionally contributed to autotrophic respiration. In addition to phenology, the greater autotrophic contribution in winter could be related to the up to 9 °C higher air than soil temperatures in March (Fig. S4). While low soil temperature likely

restricted heterotrophic respiration, vegetation was possibly metabolically active at higher air temperatures (Ferrari et al., 2018), which overall resulted in a relatively higher autotrophic respiration compared to microbial activity in cold soils. However, also in crop- and managed peatlands, total soil respiration showed more modern $\Delta^{14}CO_2$ values during winter as compared to summer. In addition, Bayesian models estimated high autotrophic contributions during winter. For these vegetation-free arable systems, this may partly result from the release of recent C during decomposition of residual plant

material from the summer growing season, which was not adequately reflected in the incubation experiments.



**4.7 Study limitations**

The present investigation comprises an extensive suite of C isotopic and supporting data from soils across diverse land-use types and ecoregions of Switzerland. Given the logistical and analytical constraints associated with [14]C measurements, we strategically focused on summer and winter sampling to capture the broadest variability in environmental conditions and plant physiological states. As a consequence of these challenges, our study provides snapshots rather than continuous records of $CO_2$ ages, source contributions, and soil respiration rates, resulting in limited spatial and temporal coverage. Despite the inherent limitations, the data reveal clear and robust patterns, offering novel and meaningful insights into C cycling pathways across different land-use types.

In winter, autotrophic respiration might have been overestimated as we likely missed the phenologically dormant state of vegetation at some sites. Furthermore, source partitioning using [14]C might overestimate autotrophic respiration in crop- and managed peatlands, as clipping of vegetation inside the chamber shortly before sampling could initiate a pulse of root respiration (Wunderlich & Borken, 2012). Two additional sources of soil respiration which remain challenging to account for are carbonate from natural or artificial (liming) rock weathering as well as the decomposition of recent plant assimilates, the latter particularly affecting crop- and managed peatlands. Despite these limitations, clear patterns have emerged from the suite of measurements undertaken as part of this study, providing the basis for a proposed conceptual framework for land-use system categorization based on the rates and [14]C signatures of soil respiration.

**5 Conclusions**

Our results, revealing that the significant differences in soil respiration rates, SOC decomposability, ages, and source contributions of soil respired $CO_2$ allowed us to categorize land-use types according to their C cycling pathways. We identified the following categories of systems:

(i) Temperate grasslands are **high-throughput systems,** which release modern $^{14}CO_2$ at a high rate, driven by high autotrophic contributions and high below ground C inputs of recent plant material.

(ii) Alpine grasslands are **retarding systems**, where high SOC decomposability in conjunction with decadal (topsoil) up to centennial (subsoil) heterotrophic $CO_2$ release indicates reduced cycling rates due to cooler climatic conditions. The accumulation of decomposable old SOC makes alpine grasslands highly vulnerable to rapid climate warming or disturbances.

(iii) Forests are **preserving systems**, where low *in situ* release of decadal-old $CO_2$ and a dominance of heterotrophic respiration indicate stabilization and delayed release of carbon.

(iv) Croplands are **destabilized C-depleted systems**, where reduced C inputs and tillage result in a depletion of SOC stocks. This is reflected in low SOC decomposability and the heterotrophic release of $CO_2$ that is several hundred years old. The remaining old SOM is rather recalcitrant and leads to a low contribution of heterotrophic respiration.



(v)     Managed peatlands are **destabilized hotspots** of old $CO_2$ release due to the aeration of formerly water-logged peat soils and the decomposition of once-preserved SOC.

Overall, we propose that measuring both the rates and ages of $CO_2$ from *in situ* soil respiration and soil incubations provides valuable insights into C cycling pathways in different land-use types, as well as their capacity to retain SOC and/or their vulnerability to SOC destabilization. This framework is also helpful to associate changes in C cycling with environmental disturbances (i.e., land-use and climate change). We further suggest that the relationship between rates and ages of soil-respired $CO_2$ is a robust indicator of C retention and destabilization and could help to categorize ecosystems along the trajectory from

natural to disturbed systems on a global scale.

## Code and data availability

The code and underlying data supporting this study will be submitted to the Zenodo repository upon acceptance of the manuscript.

## Author Contribution

**L. I. M.:** conceptualisation, data curation, formal analysis, investigation, methodology, project administration, visualisation, writing – original draft preparation, writing – review and editing. **D. G.:** methodology, writing – review and editing. **S. T.:** investigation, formal analysis. **A. U.:** investigation, methodology, writing – review and editing. **A. S. B.:** methodology, formal analysis, writing – review and editing. **M. M. D.:** investigation, writing – review and editing. **C. M.:** investigation, writing – review and editing. **L. W.:** methodology, resources, writing – review and editing. **P. G.:** methodology, resources. **N. H.:**

methodology, resources. **M. E.:** investigation, writing – review and editing. **A. K.:** investigation. **J. L.:** investigation, writing – review and editing. **T. I. E.:** funding acquisition, resources, supervision, writing – review and editing. **F. H.:** conceptualisation, funding acquisition, resources, supervision, writing – original draft preparation, writing – review and editing.

## Competing interest

At least one of the (co-)authors is a member of the editorial board of Biogeosciences.

## Acknowledgements

We thank Michael Leonardo di Gallo, Logan James Banner, Andrin Bieri, Nik Wirz, Clara Juliette Gund, David Schweizer, Thomas Laemmel, Niek Abram ten Cate, and Dennis Christopher Handte for their support during field work. We further thank Alois Zürcher, Daniel Christen, Daniel Wasner, David Schweizer, Jonathan Frei, Roger Köchli, and Marco Walser for their



assistance with the laboratory work. Further thanks to André Albrecht and Urs Ramsperger for their support during sample preparation and measurements of [14]C contents and to Alessandro Schlumpf and Ursula Graf for their support during sample preparation and measurements of [13]C contents. Special thanks to our colleagues Richard Peters, Thomas Guillaume, Juliane Hirte, Raphaël Wittwer, Sonja Paul Marit, Matthias Volk, Yi Wang, and Nina Buchmann for providing information and access to sampling sites. We further thank Mirko Städler, René Haselbacher, Peter Röthlisberger, and Matthias Gyger for providing

their fields for our sampling. In addition, we thank Claudia Guidi and Katrin Meusburger for their support and scientific input during data analysis.

**Financial support**

Financial support was provided by the Radiocarbon Inventories of Switzerland project 193770 funded by the Swiss National Science Foundation.

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
