# Peer review of "Conceptualising carbon cycling pathways across different land-use types based on rates and ages of soil-respired CO2"

_EGUsphere, 2025_

## Author Response (AR2)

**General response:**

Dear Editor and Referees,

Thank you for accepting our manuscript for publication in *Biogeosciences* following corrections. As requested, we have revised the manuscript and addressed the remaining comments raised by both the referees and the editor.

The main revisions include the preparation of a revised Conclusions section and clarification of several methodological aspects. We believe that addressing these remaining points has further improved the clarity and quality of the manuscript.

Please note that the data supporting this study, together with the codes used for the Bayesian mixing model, are now publicly available in the open-access Zenodo repository under DOI: https://doi.org/10.5281/zenodo.18067939. Access to the data is currently restricted but will be made open upon acceptance of the manuscript.

Detailed responses to the editor's comments are provided below. Line references in the responses refer to the revised version of the manuscript.

Kind regards,
Luisa Minich

**Specific responses to editor comments:**

L20 I would eliminate 'July/August and March', and rather explain the rationale of the sampling time. You should mention in the abstract the incubation experiment.
Response: Thank you for this comment. We adjusted the abstract accordingly.

L24-33 I'm not convinced about the structure of the main findings in the abstract. I'd keep it simple, as you did in the second paragraph of the Discussion.
Response: Thank you for this comment. We simplified the C cycling categories in the abstract.

L138-39 be more specific about the choice of the sampling campaign period
Response: Thank you for this comment. However, we would like to point out, that we already provide the rationale for the choice of the sampling period in the current version of the manuscript (lines 136-138): *"These months were selected to capture the greatest variability in the soil environmental conditions while avoiding snow cover and melt. Severe climatic conditions precluded a sampling in March at the three alpine grassland sites."*

L156 only at forest sites?
Response: Yes, the replicated sampling was only performed for the forest sites Hölstein and Pfynwald due to a limited capacity of $^{14}C$ measurements. We chose forests for the replicated sampling as we expected the highest variability in soil respiration across landuse types, due to their greater diversity of plant species and generally more heterogeneous distribution of the vegetation cover.

L182 only heterotrophic endmembers? What about the rest?

Response: We determined isotopic signatures of heterotrophic as well as of autotrophic endmembers, as denoted in the methods section (line 179-180): *"We determined isotopic signatures of autotrophic and heterotrophic endmembers by conducting short-term root and soil incubations."*

L186-187 you say that 'δ13C value of excised roots can slightly change after a few minutes' and then that 'Roots were incubated overnight'. Were the results reliable?

Response: Thank you for this comment. We are confident that our results are reliable and that the root incubation approach is appropriate. Previous work has shown that the $\delta^{13}C$ value of excised roots changes only slightly over short time periods (Midwood et al., 2006). To minimize any potential changes, roots were incubated immediately in the field after excision. An overnight incubation was necessary to accumulate sufficient $CO_2$ for subsequent isotopic analyses. This approach has been successfully applied in earlier studies using both $^{13}C$ and $^{14}C$ signatures for source partitioning of soil respiration (e.g., Schuur et al., 2006). Importantly, the $\delta^{13}C$ signature of autotrophic respiration was consistently more depleted than that of heterotrophic respiration across all samples, indicating that distinct and robust isotopic endmembers were captured.

L223-225 Revise meaning of this sentence. It is not clear.

Response: Thank you for this remark. We revised the sentence to *"Although we estimated carbonate contributions with certain limitations (Supplement S7), we were unable to accurately correct the $^{14}C$ isotopic signature of heterotrophically respired $CO_2$ because the isotopic signatures of endmembers needed for this correction (i.e., carbonate and SOC-derived $CO_2$) could not be sufficiently constrained."*

L288 What do you mean by 'modern levels'?

Response: Thank you for pointing this out. The wording "modern levels" indicates that the $^{14}C$ value of soil-respired $CO_2$ is close to contemporary atmospheric levels and thus young. We rephrased this sentence to be more explicit: *"The in situ soil-respired $^{14}CO_2$ in croplands and grasslands exhibited modern values, close to the contemporary atmospheric $^{14}CO_2$, corresponding to ages < 10 years (Table 2)."*

L301-301 '...reduced respiratory activity and atmospheric mixing... ' - complete the sentence

Response: Thank you for this comment. We rephrased this sentence to be more concise: *"...generally related to reduced respiratory activity and seasonal variation in atmospheric transport of $CO_2$ (Schuur et al., 2016)."*

L366 What do you mean by 'plant-soil' regime?

Response: Thank you for pointing this out. By *"plant–soil regime"* we refer to C cycling within the coupled plant–soil system, encompassing both plant inputs and soil

processes. Total soil respiration integrates $CO_2$ released from heterotrophic (microbial) respiration and autotrophic (root) respiration. We therefore used this term to indicate that differences in C cycling among land-use types are not restricted to soil processes alone but also involve plant-related pathways, as reflected by differences in $^{14}C$ signatures of $CO_2$ derived from autotrophic respiration.

L494 Add all study limitations including those linked to the incubation experiment
Response: Thank you for this comment. We believe that the major study limitations are already addressed in section 4.7 (Study limitations). In particular, limitations associated with the incubation experiment include the challenge of accounting for additional $CO_2$ sources, such as carbonate weathering (from natural or liming-derived carbonates) and the decomposition of recent plant inputs. These potential contributions are discussed in lines 508-512. We have now revised this paragraph to more explicitly link these processes to the incubation experiment:

*"Two additional sources of soil respiration that may contribute to $CO_2$ sampled during soil incubations remain challenging to account for: carbonate weathering from natural or artificial (liming) sources, and the decomposition of recent plant assimilates, the latter particularly affecting croplands and managed peatlands."*

A further, minor constraint of the incubation approach is the potential disruption of soil aggregates caused by 4 mm-sieving prior to incubation. This limitation is already acknowledged in the methods section (section 2.3, lines 170-173). Sieving was necessary to remove roots, which would otherwise have exerted a likely stronger influence on the isotopic signatures than the partial exposure of SOC resulting from aggregate disruption. Given its relatively minor impact compared to the limitations discussed above, we consider it appropriate to retain this point in the methods section rather than expanding it in the discussion.

L515 Avoid repeating what already said in the abstract and focus on the implications of your study
Response: Thank you for this comment. Since the similarity of the conclusions with the abstract was also pointed out by Referee 2, we present a new version of the conclusions in the revised manuscript.

Figure legends and table captions need to be self-standing (where? when? why?) and contain all acronym definitions. Identify error bars (SE? SD?).
Response: Thank you for this remark. We revised figure captions when necessary.

Table 1 Opt for different symbols other than repeating '*' to explain the origin of climate variables; it might be confused with significance values. Shorten each of the explanations by eliminating redundant information (e.g. MeteoSwiss station: Weissflühhorn (2690 Hm))
Response: Thank you for this comment. We changed the symbols of footnotes from "*,

**, …, *****" to "a), b), c)". Further, we shortened and aggregated the footnote explanations.

Figure 4 In the above panel SE / SD bars are missing.

Response: Thank you for this comment. The upper panel of Figure 4 shows cumulative SOC decomposability for the organic layer, topsoil, and subsoil across sites. For each site and depth, we incubated a single composite sample, resulting in only one cumulative SOC decomposability per site and depth layer. Therefore, SE or SD bars cannot be shown for this panel.